# Identifying Weak Adhesion in Single-Lap Joints Using Lamb Wave Data and Artificial Intelligence Algorithms

**Gabriel M. F. Ramalho** [1,*] , **António M. Lopes** [1,2,*] , **Ricardo J. C. Carbas** [1,2] and **Lucas F. M. Da Silva** [1,2]

1 Faculty of Engineering, University of Porto, 4200-465 Porto, Portugal
2 INEGI—Institute of Science and Innovation in Mechanical and Industrial Engineering, 4200-465 Porto, Portugal
* Correspondence: gmframalho@gmail.com (G.M.F.R.); aml@fe.up.pt (A.M.L.)

**Abstract:** In the last few years, the application of adhesive joints has grown significantly. Adhesive joints are often affected by a specific type of defect known as weak adhesion, which can only be effectively detected through destructive tests. In this paper, we propose nondestructive testing techniques to detect weak adhesion. These are based on Lamb wave (LW) data and artificial intelligence algorithms. A dataset consisting of simulated LW time series extracted from single-lap joints (SLJs) subjected to multiple levels of weak adhesion was generated. The raw time series were pre-processed to avoid numerical saturation and to remove outliers. The processed data were then used as the input to different artificial intelligence algorithms, namely feedforward neural networks (FNNs), long short-term memory (LSTM) networks, gated recurrent unit (GRU) networks, and convolutional neural networks (CNNs), for their training and testing. The results showed that all algorithms were capable of detecting up to 20 different levels of weak adhesion in SLJs, with an overall accuracy between 97% and 99%. Regarding the training time, the FNN emerged as the most-appropriate. On the other hand, the GRU showed overall faster learning, being able to converge in less than 50 epochs. Therefore, the FNN and GRU presented the best accuracy and had relatively acceptable convergence times, making them the most-suitable choices. The proposed approach constitutes a new framework allowing the creation of standardized data and optimal algorithm selection for further work on nondestructive damage detection and localization in adhesive joints.

**Keywords:** artificial intelligence; adhesive joints; machine learning; Lamb wave

## 1. Introduction

In recent years, there has been an increase in the use of adhesive joints in many industries. They are attractive due to their ability to bond different classes of materials, such as metals and composites, while uniformly distributing stress and being lighter than conventional joining methods, namely rivets [1–3]. Different types of adhesive joints can be manufactured, the single-lap joint (SLJ) being one of the most-used, with applications ranging from the automotive to the aeronautical industries [4]. Although the adoption of adhesive joints is widespread, their use in critical primary structures is still limited due to their inability to meet the needed strict certification and inspection criteria [5]. Existing standards require that, for any joining mechanism, there must exist reliable and robust nondestructive testing (NDT) methods for damage detection. Current NDT methods have strong limitations with adhesive joints, and certain types of damage cannot be effectively detected [5–7], as is the case of weak adhesion. Weak adhesion is characterized by a reduction of at least 20% of the maximum adhesive strength of the joint, without presenting any discernible voids, cracks, or other visible alterations in the structure. Weak adhesion is normally caused by contamination between the adhesive and the substrate and is commonly due to the presence of oil or dust or improper surface preparation for the desired adhesive [8].

Some NDT methods have been proposed for detecting weak adhesion. The one that has recently shown the most-promising results involves Lamb waves (LWs). LWs are a form of elastic disturbance that propagates trough thin plates with shallow to no curvature and possess the capacity to travel long distances without large signal attenuation [9,10]. These properties come with the disadvantage that the raw signals are extremely complex, and depending on the frequencies and materials used, they are multimodal, which further complicates any required type of analysis [11]. This complexity can be mitigated by choosing appropriate methods to evaluate and process the LW data. One such option is the use of machine learning methods applied to the time series, as they allow a hands-off approach for detecting and classifying any damage present in the bonded structures. For using machine learning, large datasets are often needed for training and testing the algorithms. Creating large datasets experimentally is costly or even impracticable. Therefore, developing numerical models of the SLJ and, subsequently, validating them with experimental data reduce the overall burden. An added benefit is that any large volumes of data can be obtained to develop machine learning algorithms.

This paper proposes an NDT technique based on LW data and artificial intelligence algorithms for detecting weak adhesion in SLJs. Initially, a dataset was generated by simulating LWs propagating in an SLJ subjected to multiple levels of weak adhesion. This was accomplished by a finite-element model (FEM) developed in the software Abaqus (Dassault Systemes, Vélizy-Villacoublay, France), which generated over 1000 variations of weak adhesion at the SLJ. The numerical data were validated with experimental tests, for the sake of reliability, thus forming the base of the present work. The raw data were preprocessed to remove outliers and generate normalized time series. These processed data were then used in different artificial intelligence algorithms, namely feedforward neural networks (FNNs), long short-term memory (LSTM) networks, gated recurrent unit (GRU) networks, and convolutional neural networks (CNNs). These data-driven algorithms were chosen as they are widely used for time series forecasting, while still being commonly used in other non-time related datasets. The algorithms were adjusted and compared in terms of accuracy and training burden. The four methods for LW data processing allowed not only accurately detecting up to 20 different levels of weak adhesion in SLJs, but also choosing an efficient training method for the problem at hand. The proposed framework allows the correlation between the methods' parameters, accuracy, and training time and can be further used with different data and algorithms.

This paper is divided into sections as follows: Section 2 presents a literature review on the subjects addressed in the paper, namely LWs, adhesive joints, and defects. Section 3 introduces each artificial intelligence method adopted in the paper. Section 4 describes the SLJ model and the generation of the dataset. Section 5 presents the algorithms' adjustments and discusses the obtained results. Finally, Section 6 summarizes the main conclusions.

## 2. Preliminary Concepts

This section presents a model for LWs and introduces adhesive joints and their possible defects.

### 2.1. Lamb Waves

An LW is a type of elastic guided wave that moves through a solid medium, exhibiting pressure (P) and shear (S) propagation modes. When applied to a medium that has boundary conditions, reflection emerges and creates complex patterns. LWs present a particularly interesting property, which is their capacity to travel long distances and even pass through multiple materials with small attenuation [12,13]. When LWs interact with a defect in the material, they alter their wave pattern, and thus, they are sensitive enough to detect, and even discern between, the damage type, size, and location [14]. When propagating in a thin plate, an LW presents two fundamental modes, namely the symmetric (Sn) and the

anti-symmetric (An) modes. These modes occur in relation to the mid-plane of the plate. The vector of displacement of these waves can be analytically described by

$$\mu \nabla^2 \mathbf{u} + (\lambda + \mu)\nabla\nabla \bullet \mathbf{u} = \rho \frac{\partial^2 \mathbf{u}}{\partial t^2},$$ (1)

where $\mu$ and $\lambda$ stand for the Lamé constants, which are two material-dependent quantities that come from the elastic stress–strain relationships, $\rho$ is the material's mass density, and the vector $\mathbf{u}$ represents the displacement, such that

$$\mathbf{u} = \nabla\Phi + \nabla \times \Psi,$$ (2)

with $\Phi$ and $\Psi$ denoting potentials.

These equations can be divided into the transverse and longitudinal displacements and be written, in turn, by defining the longitudinal (L) wave speed $c_L^2 = (\lambda + 2\mu)/\rho$ and the transverse (T) wave speed $c_T^2 = \mu/\rho$, yielding:

$$\frac{\partial^2 \Phi}{\partial x^2} + \frac{\partial^2 \Phi}{\partial y^2} + \frac{\omega^2}{c_L^2}\Phi = 0,$$

$$\frac{\partial^2 \Psi}{\partial x^2} + \frac{\partial^2 \Psi}{\partial y^2} + \frac{\omega^2}{c_T^2}\Psi = 0,$$ (3)

where $\omega$ stands for the angular frequency.

One way to visualize LWs is through their dispersion curves, where it is possible to depict the phase alteration depending on the frequency [15–18]. In the case of damage detection applications, the dispersion must be kept to a minimum, together with the attenuation, while still maximizing the sensitivity. This can be achieved by using a windowed tone burst as an excitation signal, as is the case of a Hann window. Several possible actuators can be employed, namely piezoelectric, due to their versatility, lightweight, and low cost. These elements have the advantage of being used also as sensors and possess a wide bandwidth, which can allow for multiple frequencies to be studied [14,19].

### 2.2. Adhesive Joints and Defects

An adhesive is a substance that has the capacity to join two materials together, independently of their (dis)similarity, giving them the ability to resist any form of separation [20]. When an adhesive connects two separate parts, an adhesive joint is formed. This concept can be further expanded by classifying the parts as substrates, before the bonding process, and adherents, after the creation of the joint [20]. Furthermore, the adhesive layer in contact with the adherents is called an interface, also known as the boundary layer [6].

When comparing adhesive joints with traditional fastening methods, their benefits include a more uniform distribution of stress along the bonded area, which yields good resistance to dynamic solicitation and excellent load transmission, as well as higher stiffness. Due to not using bulky fasteners, there is also a large reduction of the weight of the structure and, consequently, a minimization of the cost. However, these benefits also come with many disadvantages, which hinder the widespread application of bonded primary structures. Indeed, a major drawback is the absence of effective NDT techniques to detect damage [6,20]. The NDT methods need to be robust because, even if an adhesive is adequate for an application, many variables can alter the overall quality of the bonded joint and lead to premature mechanical failure. Common problems affecting adhesive joints are the lack of proper surface finishing, an incomplete or altered curing cycle, high humidity, or a degraded chemical composition caused by improper handling or storing of an adhesive [21]. These perturbations, when verified together, can cause a defective part, but there are cases when only one of these issues occurs. This can lead to damage types like those illustrated in Figure 1. These defects can occur in several places in the bond, such as cohesively within the adhesive, inter-facially between the adherend and adhesive,

or segmented throughout the bonded joint. Often, these are voids, debonds, porosity, or cracks within the adhesive layer. A few of these defects are detectable by certain NDT methods, but there is one defect in particular that still cannot be effectively detected by NDT, specifically weak adhesion. This defect consists of an inter-facial defect that reduces the maximum strength of an adhesive joint by at least 20% and is extremely hard to detect by conventional methods, as there is no gap or alteration in the structure; just a small reduction in the overall surface adhesion exists.

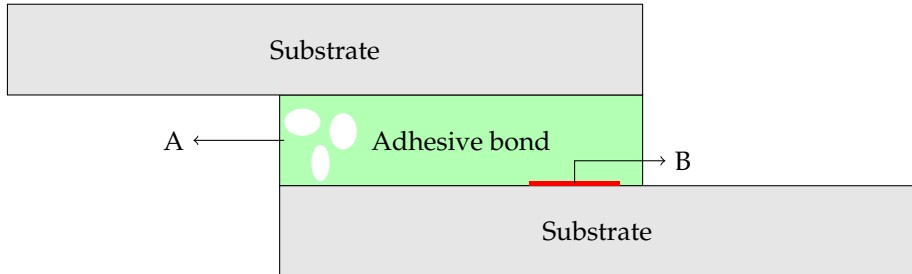

**Figure 1.** Representation of common defects in an SLJ, with "A" illustrating voids in the adhesive and "B" indicating weak adhesion.

## 3. Artificial Intelligence Algorithms

This section introduces the machine learning algorithms that were used in this paper.

### 3.1. Feedforward Neural Networks

The multi-layer FNN has been extensively used in multiple applications, starting with the work [22], where backpropagation algorithms were first used to adjust the weights in the hidden layers. In an FNN, the elements that process the data, also known as neurons, are structured in multiple successive layers that go from the input to the output (Figure 2). These layers are all interconnected in such a way that all elements in a layer have their output connected to every element in the next layer. Associated with each connection, there is a value, called the weight, which is adjusted during the learning phase [23,24]. Each neuron of a layer applies the associated weight and utilizes a nonlinear function to transform the data that pass through it. A commonly used function is the sigmoid function [25]:

$$g(h) = \frac{1}{1 + e^{-h}}. \tag{4}$$

where $h$ denotes a dimensionless variable.

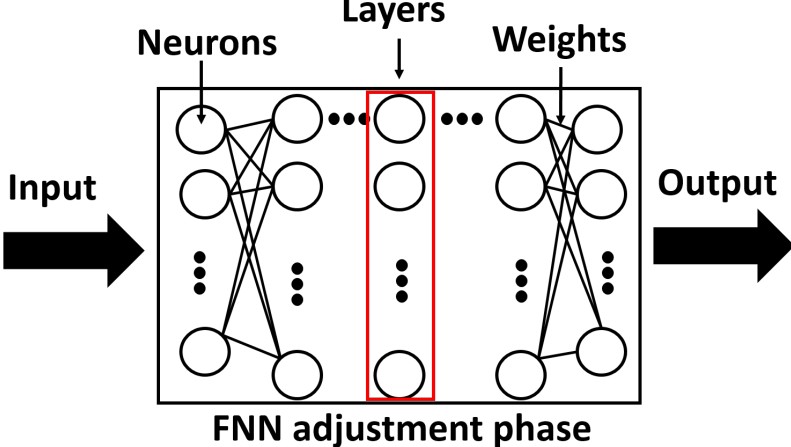

**Figure 2.** Fundamental architecture of an FNN with the interaction of each component represented.

The general and simplified intent behind an FNN is to reduce the error between the processed data and target values previously known. This is performed by passing the data through the neurons from the input to the output and, then, comparing the values with the training data previously saved and known. The error that is obtained is used to determine a performance function, which is continuously differentiable and correlates all the weights present from the output to the input layer. Two examples of such performance functions that can be used are the gradient-descent-based algorithms and the mean-squared error (MSE). These functions have weights as variables and are minimized by altering them [25]. The FNN model has been proven to be successful in modeling complex problems in a large range of areas, from image compression to forecasting and signal prediction [22]. However, this model presents the disadvantage that all elements in each layer are interconnected to the next and previous layers. This requires a careful analysis of the problem when mapping the data, as well as caution relative to the number of layers and elements used. The higher the number of layers, the larger the datasets required and the more computationally intensive the process will be. Thus, it is preferable to implement an FNN with a small number of layers and elements [22].

*3.2. Recurrent Neural Network*

The RNN is a type of artificial neural network that differentiates itself by "memorizing" information that can be later used in new data. This is performed by storing the output activation from the layers and applying them together when feeding a new input to the algorithm [26]. Although RNNs are powerful and have been applied to many problems, they suffer from a reduction or "explosion" in the gradient during backpropagation. This, in turn, does not allow for the layers closer to the input to have their weights altered or altered to such a degree that they make others irrelevant and, thus, not contribute positively to the solution [27,28]. To mitigate this problem, a few algorithms, such as LSTM and the GRU, which contain different gating units, have been created.

3.2.1. Long Short-Term Memory

To mitigate the vanishing gradient problem, LSTM utilizes an input gate and a forget gate, as shown in Figure 3 (circled in green and yellow colors, respectively), to create a constant error carousel (CEC) [29,30]. The CEC works by enforcing a constant error flow within a special cell controlled by the gate units, which learn when to grant access (circled in red color) [31]. The various gates used in the CEC can be described as

$$f_t = \sigma(W_f[h_{t-1}, x_t] + b_f), \tag{5}$$

$$i_t = \sigma(W_i[h_{t-1}, x_t] + b_i), \tag{6}$$

$$o_t = \sigma(W_o[h_{t-1}, x_t] + b_o), \tag{7}$$

while the equations for the candidate state, cell state, and final output are

$$c'_t = \tanh(W_c[h_{t-1}, x_t] + b_c), \tag{8}$$

$$c_t = f_t c_{t-1} + i_t c'_t, \tag{9}$$

$$h_t = o_t \cdot \tanh(c_t). \tag{10}$$

In these equations, $f_t$ is the forget gate, $i_t$ is the input gate, $o_t$ is the output gate, $c_t$ is the cell state, $c'_t$ is the candidate state, and $h_t$ is the final output. Furthermore, $W_x$ represents applied weights; $h_{t-1}$ are the outputs of the previous LSTM block; $x_t$ are the inputs at the current timestamps; $b_x$ are the biases for the gates. These gates can be easily visualized in Figure 3, where the components are in yellow, the layers are in dark blue, the line that divides represents a variable copied, and lines combining represent concatenation. Finally, the output gate is circled in purple color and serves as the output of the cell.

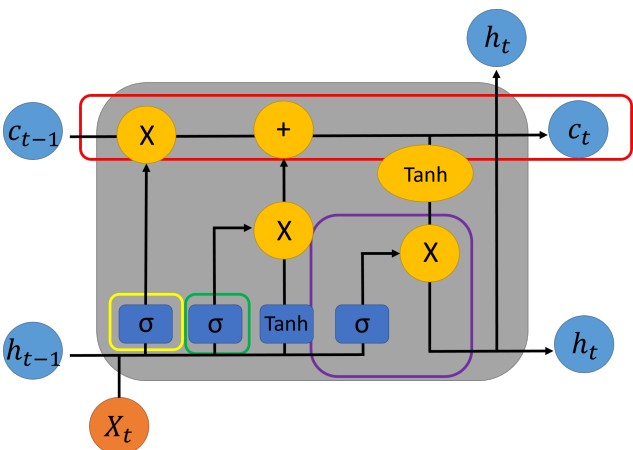

**Figure 3.** The architecture of an LSTM cell with the input cell circled in green, the forget cell in yellow, the cell state in red, and finally, the output gate in purple.

### 3.2.2. Gated Recurrent Unit

The GRU has been used in many applications, but has seen more significant results in music and speech modeling, as well as natural language processing. This method was developed with the intent of allowing each recurrent unit to alter the sequence of dependencies in an adaptive manner over time. To accomplish this, the unit was built with the following four steps, where $x_t$ is the current input, $h_{t-1}$ is the output of the previous step, $W$ and $U$ are parameter matrices for the weights, and $\sigma$ is the sigmoid function [29,32]:

1.  The first step is an update gate, as shown in Figure 4 (circled in green color), which is used to determine how much information will be updated in the unit. This gate is denoted as $z_t^j$ and calculated by:

$$z_t^j = \sigma(W_z x_t + U_z h_{t-1})^j.$$

2.  The next step is a reset gate, seen in Figure 4, circled in purple, denoted as $r_t^j$ and calculated by:

$$r_t^j = \sigma(W_r x_t + U_r h_{t-1})^j.$$

    This gate is used to determine how much information needs to be forgotten. When the value of this gate is close to zero, it determines that the *j*-th information should be forgotten in the current memory. Any value close to one means that the data will be preserved.

3.  The third step is to determine the current memory. This gate is denoted as $h_t^j$ in the following expression and, in Figure 4, seen as $h_t$.

$$h_t' = \tanh(W_h x_t + r_t \cdot U_h h_{t-1}))^j.$$

    This gate also uses the Hadamard product to decide how much of the hidden state of the content should be forgotten.

4.  Finally, the last step is to determine the information to be stored in the hidden layer at the current iteration that will be passed onto the next cell. This step is denoted as $h_t$ in the following equation and, in Figure 4, is denoted as $h_{t-1}$:

$$h_t = (1 - z_t^j)h_t^j + z_t^j \widetilde{h}_t^j.$$

These steps can all be easily visualized in Figure 4.

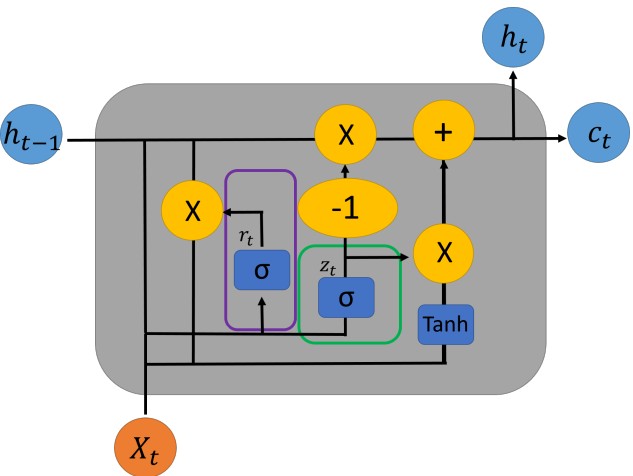

**Figure 4.** Architecture of a GRU cell with the reset gate circled in purple and the update gate in green.

### 3.3. Convolutional Neural Network

Overall, the CNN can be seen as having a multi-layer feedforward structure and, normally, utilizes other types of layers as a complement to the convolutional ones. The main characteristics that define a CNN are the representation of the input layer and the way these input elements are processed in the CNN layer, which allows for significant advantages while creating a few problems. The first layer used is normally seen as a grid that can be compared to 2D images or pixels (or even a signal in the time domain arranged in an $n \times 1$ column vector), as seen in Figure 5.

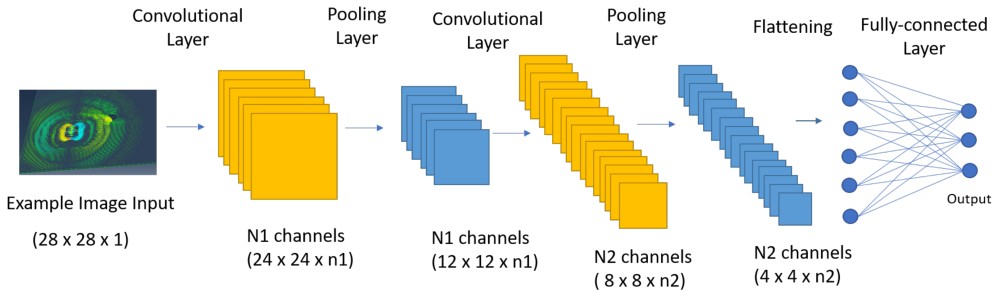

**Figure 5.** Architecture of a CNN comprising convolutional and pooling layers with a final flattening layer. The dimensions of each layer are represented under each one.

Another characteristic that defines a CNN is a convolutional operation called the kernel, which is applied between the weights and the input CNN layer. This is performed to provide at the output of the layer a feature map [33,34]. When evaluating the inputs and the kernels used in a convolutional layer as square $n \times n$ arrays, the size of the output layer can be calculated using

$$n_{out} = \left\lceil \frac{n_{in} + 2p - k}{s} \right\rceil + 1, \tag{11}$$

where the width or height of the output and input arrays is represented by $n_{out}$ and $n_{in}$, respectively. Finally, two other parameters complement these layers: the convolutional padding size is a parameter that is used when it becomes adequate to include additional information on the border arrays, while the sliding increment of the kernels over the input array is the stride parameter. These can be seen as elaborated in the following equations:

$$s(t) = \int x(a)w(t-a)da, \tag{12}$$

$$s(t) = (x * w)(t), \tag{13}$$

$$s(t) = (x * w)(t) = \sum_{a=-\infty}^{\infty} x(a)w(t-a),  \qquad (14)$$

where $n_{out}$ represents the number of output features obtained from the convolution, $n_{in}$ is the number of input features, $p$ is the convolutional padding size, $k$ stands for the kernel size, and $s$ is the stride size to be used. One way to better visualize these kernels is to see them as filters that go through the input values, using a constant weight to identify specific features in the input data. These values are refined during the learning phase. Using a CNN instead of an FNN allows for a smaller increase in the number of variables and parameters when setting more layers, as the weights are used with all inputs. This allows for more layers and a deeper learning FNN implementation. Furthermore, it allows for a way to associate values that are near one another, which can be of interest in images, time series, or any other sequential signal. Finally, the convolutional maps are normally associated with pooling stages and processed through a nonlinear rectified linear unit (RELU) function to stabilize values in specific locations [23].

### 4. Numerical Simulated Data

To fully utilize the combination of LWs and machine learning methods, it is necessary to create a large dataset that can be used to train and test the algorithms. As there is a need for a large volume of data, creating these experimentally would be labor-intensive, costly, and difficult to obtain consistently. Therefore, the alternative of using simulation data from accurate and experimentally validated numerical models has considerable advantages. Herein, a model of an SLJ composed of two aluminum substrates bonded with the epoxy-modified Nagase T-836/R-810 was adopted. This adhesive was chosen as it has great potential for industrial applications, specifically in the aeronautical and automotive sectors. However, any other epoxy-based adhesive with similar mechanical properties can be used with the same methodology. One actuator was chosen to excite the LWs, and one sensor captured the SLJ response. The specimen dimensions and the positions of the actuator and sensor can be seen in Figure 6.

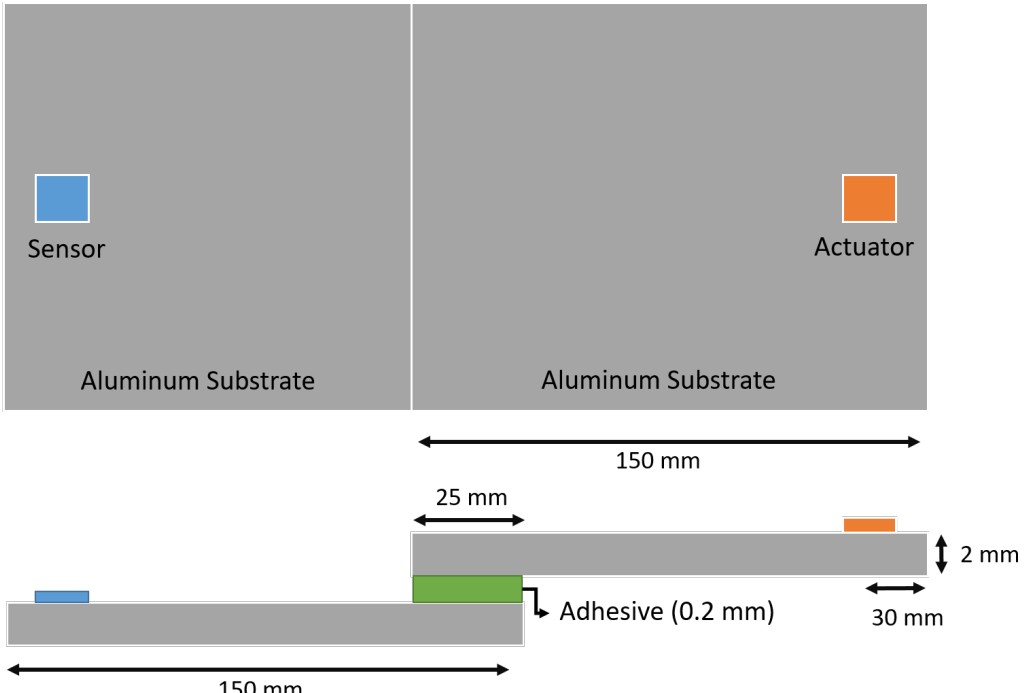

**Figure 6.** Top and side view of the adhesive joint simulation setup with the position of the sensor and actuator and the overall dimensions of each part.

The model was developed and simulated using the FEM in the software Abaqus. The two SLJ aluminum plates were modeled with a density $\rho = 2500$ kg/m$^3$, a Poisson's ratio of $\nu = 0.33$, and a Young's modulus of $E = 72.4$ GPa. Weak adhesion defects were simulated by creating a thin layer of adhesive where Young's modulus was reduced in comparison to the rest of the adhesive material. A total of 1000 cases were simulated with linearly distributed values between 600 and 270,000 kPa. For exciting the SLJ, a distributed discretized tangential force was applied to the substrate plane, as illustrated in Figure 7. A 5-cycle Hanning-windowed sinusoidal burst centered at the frequency of 100 kHz was adopted [35–38]. The amplitude of the force was adjusted in accordance with the experimental observations, by successive trials. The explicit dynamic analysis was adopted, due to its computational efficiency and because the excitation is impulsive and the time-dependent reaction needs to be captured. The total simulation time was limited to 0.5 ms, which corresponds to enough time for the LW to interact with the weak adhesion defects and arrive at the sensor [39,40]. Following a mesh convergence study, a mesh size of 1.5 mm with elements of the type C3D8R was utilized. This was performed to reduce the computational burden and, thus, the time necessary for each simulation, while still maintaining a small enough mesh to not cause any numerical divergence. This combination allowed a simulation time inferior to 20 min for each case.

## Aluminum Substrate

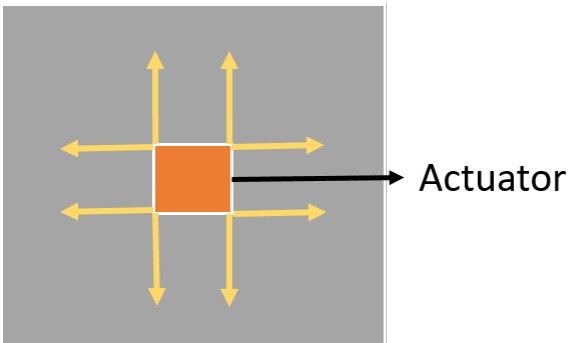

**Figure 7.** Representation of how the forces were simulated by discretizing and applying them to the edges of a square that has the same dimensions as the actuator.

Due to the explicit dynamic analysis carried out in Abaqus, the time series generated presented varying lengths, as each simulation was performed without a fixed time step. Thus, it was necessary to interpolate the data to have a fixed number of points. The value of 5000 points was chosen, this being enough to represent the LW signals well while keeping the amount of data limited.

It should be noted that the use of only one actuator/sensor pair on the SLJ is the minimum necessary to detect damage. However, it does not allow for damage localization. Indeed, this work focused only on determining the level of weak adhesion in an SLJ of reduced dimensions. For localizing damage in larger parts, extra sensors are needed, along with some triangulation methods for data processing.

## 5. Algorithm Application and Results

### 5.1. Evaluation of Components and Indicators

To use artificial neural networks, it is first necessary to describe the desired outcome of the model and, from there, determine how the task should be handled. Adhesive materials intrinsically have many variables that determine their overall bonding strength [41,42]. For a given adhesive and joint, damage can limit the joint strength to a specific interval, between no adhesion up to full-strength adhesion. Therefore, the problem at hand is considered as involving classification into a given number of weak adhesion classes, rather than a regression problem. A total of 20 classes were chosen, as this was a large enough

value to visualize any possible damage progression, while still small enough to allow for distinct classes.

With the basic concept chosen, various parameters can immediately be defined. Thus, the evaluation metric or loss function adopted was the categorical cross-entropy, and all final layers before the output contained the softmax activation function with a kernel equal to 20, that is the number of classes. Finally, the accuracy metric was chosen, as it is widely implemented in classification problems and is a simple metric to assess the different models that were used. To train the model, the data were divided into training and testing in the proportion 70% and 30%, respectively, as these percentages are classic values that work in most cases and give a good baseline to initiate the study. Finally, the chosen epoch size was 400, as the objective of this work was to properly understand each algorithm and not allow for an early stop, which could exclude important behaviors. This may cause overfitting of the training data, but this was not considered a concern during the analysis, as it was an expected outcome. Although there would be a constant of 100 epochs, each algorithm was also evaluated accordingly with the epoch at which the accuracy started to converge and how long this process took.

To give the best evaluation of each model for each type of dataset, it is necessary to assess several parameters and algorithm architectures. These parameters vary from algorithm to algorithm, and their optimization requires an extensive grid search to determine, inside the chosen grid, the best parameters. To minimize the total time expended in such an optimization, a broad trend search was conducted, where the total accuracy was compared to the time spent to reach the solution. This allowed the algorithm not only be efficient, but also faster when compared to other configurations, which can take days to train on a dataset. Initially, the parameters listed in Table 1 were chosen to start the optimization study on the FNN, as it is one of the fastest methods.

**Table 1.** Initial parameters of the FNN.

| Experimental Variables to Be Tested | | |
|---|---|---|
| **Batch Size** | **Kernel Size** | **Number of Hidden Layers** |
| 32 | 50 | 1 |
| 64 | 100 | 3 |
| 128 | 400 | 7 |

After testing these values, it was possible to see how each variable behaved. During the tests, the time spent to test a batch with a kernel size of 400 compared to a kernel size of 50 was over 12-times greater with minimal accuracy changes. Thus, all tests using kernel sizes of 400 were discarded as being too time-consuming for minimal results. In Figure 8, it is possible to see that, for an algorithm depth of one hidden layer, the batch size had a small influence on the time per training, while there was no variation in the accuracy. Therefore, a batch size of 128 was chosen, as it was the fastest in both cases with comparable results.

The number of hidden layers was tested together with multiple values of the kernel and batch sizes, and the ones with the shortest training times and highest accuracy were chosen. To compare the influence of the number of hidden layers, three tests were conducted with a constant batch size of 128, as defined previously, and a varying kernel size between 50 and 100. In Figure 9, it is possible to verify that, while there was a linear growth in time for the kernel size of 100, there was a plateau at the kernel size of 50. Therefore, it was better to not only utilize a kernel size equal to 50, but also to use more layers. This may change if the model is tested for a larger number of hidden layers or for other algorithms, but these data served as a starting point for further evaluations.

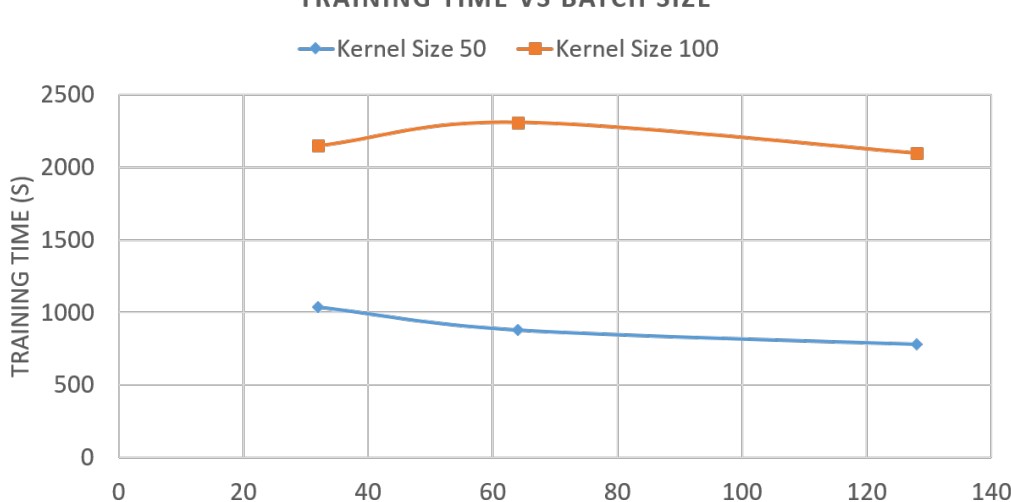

**Figure 8.** Comparison between the total training time and batch size for the FNN when fixing the kernel size to 50 (blue) and 100 (orange).

**Figure 9.** Comparison between the total training time and the number of hidden layers for the FNN when fixing the kernel size to 50 (blue) and 100 (orange).

When comparing the previous graphs and results, it is possible to conclude that the configuration of the parameters and algorithm architecture that brought the most-consistent and highest values of accuracy, while still taking the least time, was:

- Batch size: 128;
- Number of hidden layers: seven;
- Kernel size: 50.

Therefore, the chosen configuration for the FNN had a total of four dense layers, as shown in Figure 10.

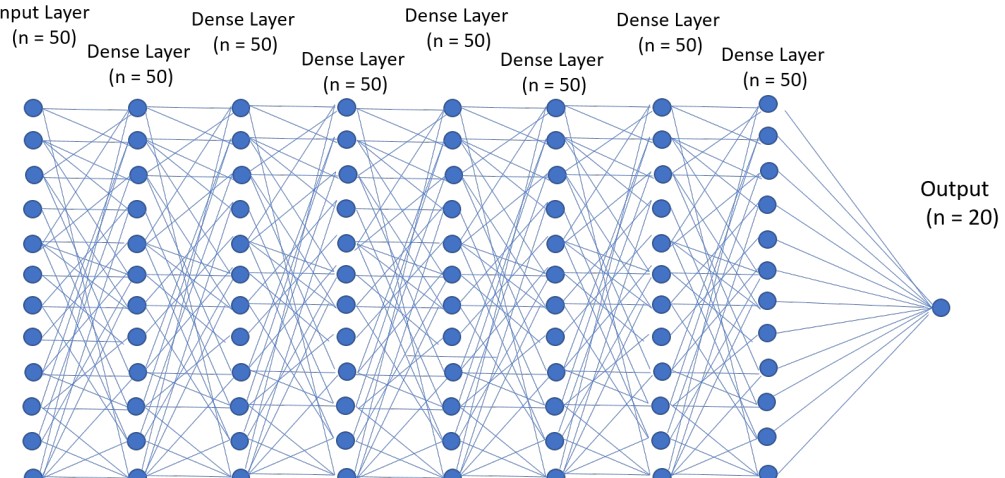

**Figure 10.** Graphic representation of the FNN configuration, chosen after the optimization process, with a total of 7 dense hidden layers.

Although the batch size can be defined for all algorithms, every one has specific characteristics that need to be altered individually. One example of this concerns the CNN, which not only has the kernel size, but also has filters and utilizes max pooling layers, which have their own individual parameters, namely the pool size. In these cases, the base parameters are maintained as much as possible, while each unique parameter must be individually studied.

When optimizing the two RNN algorithms, the main obstacle was related to the training time. This was not only due to a large number of inputs, but also to the very nature of the algorithms, in which each neuron had a recurring unit, which further necessitated calculations and, thus, required additional computational power. Due to these restrictions, the optimization of the RNNs was limited to creating a model that requires a training time similar to that of the FNN and CNN. This optimization led to a simple algorithm that had only two recurring layers, each one with a kernel size equal to 10. The batch size was maintained from the previous analysis at the value of 128. Therefore, the chosen configuration for the GRU had a total of two recurrent GRU layers, as shown in Figure 11, and for the LSTM, there was a total of four dense layers, as shown in Figure 12.

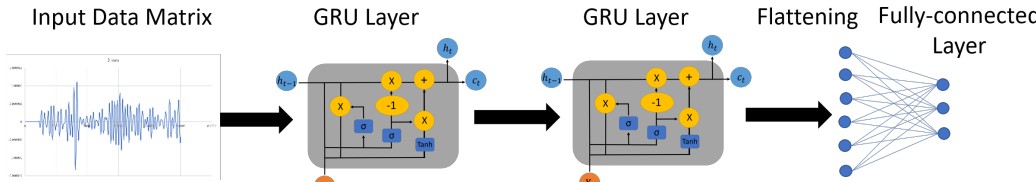

**Figure 11.** Graphic representation of the GRU's configuration, chosen after the optimization process, with two layers.

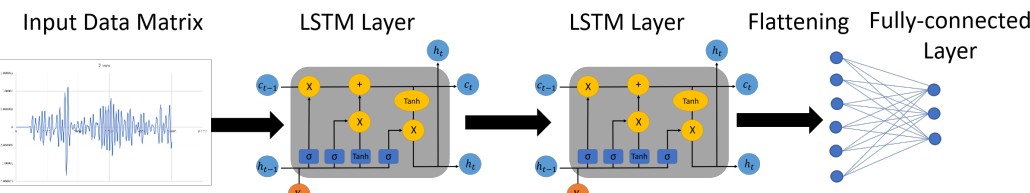

**Figure 12.** Graphic representation of the LSTM's configuration, chosen after the optimization process, with two layers.

In comparison to the RNNs, the CNN algorithm did not have the training time as a limiting factor, and thus, it was possible to test multiple configurations, as done with the FNN. The first analysis conducted was to determine the batch size that would be used for the CNN. It is possible to see in Figure 13 that the overall batch size did not influence the training time greatly, while the size of the kernel did. Therefore, the chosen batch size was 64, as it is an average value that could be used as a base for the next variable testing. It is important to note that, although not explicitly shown in the following figures, the first criterion when choosing the variables was the achieved overall accuracy, and only after confirming that the differences in the accuracy were small, the time criterion was used.

**Figure 13.** Comparison between the total training time and the number of hidden layers for the CNN when fixing the kernel size to 5 (blue) and 10 (orange).

With the batch size chosen, the next variable studied was the kernel size. It was already shown in Figure 13 that the kernel size played an important role in the training time, and this becomes more apparent in Figure 14, where it is possible to see that, with low pooling sizes, the training time grew significantly. Therefore, to minimize the training, the chosen kernel size was five.

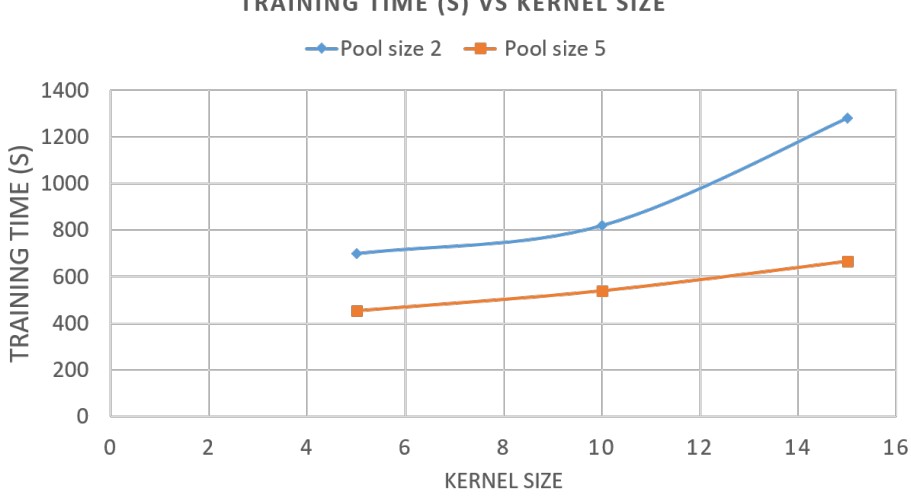

**Figure 14.** Comparison between the total training time and the kernel size for the CNN when fixing the pool size to 2 (blue) and 5 (orange).

Although Figure 15 shows that the fastest training cycles occurred at a pool size of 10 and a filter of 50, the chosen values for these were slightly different. To choose the pooling and filter size, the accuracy criterion was used instead of the time, as the values

obtained from a pooling size of two and a filter size of 50 were slightly larger than the other values.

**TRAINING TIME (S) VS POOL SIZE**

**Figure 15.** Comparison between the total training time and the pool size for the CNN when fixing the filter size to 50 (blue) and 100 (orange).

Therefore, the chosen configuration for the CNN had a total of two convolutional and two pooling layers, as shown in Figure 16.

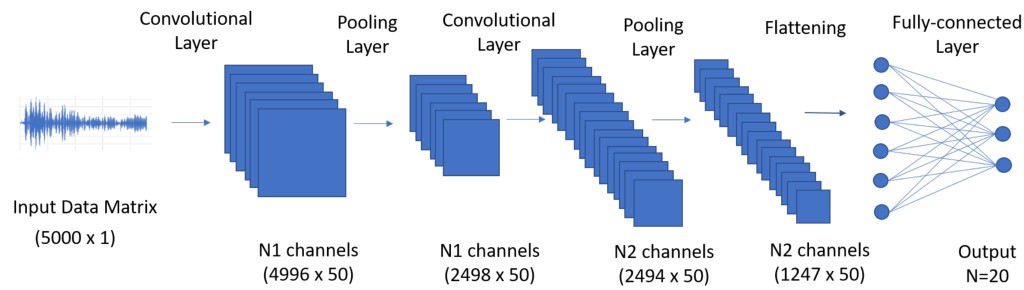

**Figure 16.** Chosen architecture for the CNN, with two convolutional and two pooling layers with the total size of each layer represented under the graphical representation.

### 5.2. Results and Discussion

With all models' architecture and parameters chosen, the algorithms were trained for 400 epochs. As stated previously, this was performed to fully understand the behavior of each algorithm. As expected, after model optimization, the algorithms quickly converged and had the lowest loss values before 100 epochs, as seen in Figures 17–20. For the FNN, at about 250 epochs, the validation loss function started to grow, indicating that the algorithm was overfitting the data. While this showed that the FNN was more prone to overfitting compared to the other algorithms, the quick convergence mitigated the problem and allowed the algorithm to be suitable for the problem at hand. For the other algorithms, no overfitting occurred, even after 400 epochs, showing that they were more robust.

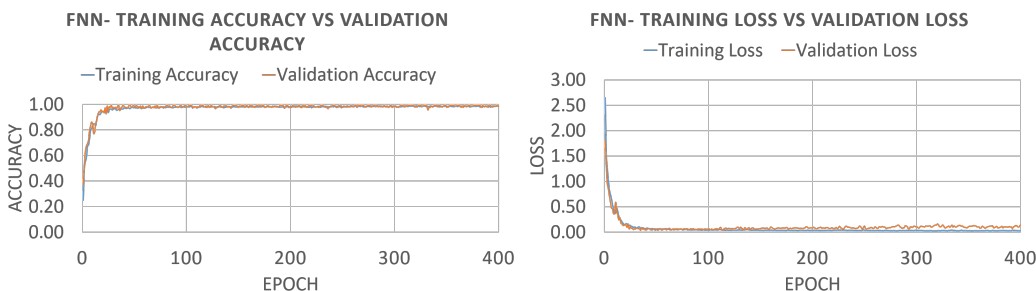

**Figure 17.** Comparison between the loss and accuracy for the FNN algorithm in both training and validation cycles.

Relative to the FNN, the CNN presented little variation in the graphs, with a smooth decrease in loss and a steady increase in accuracy.

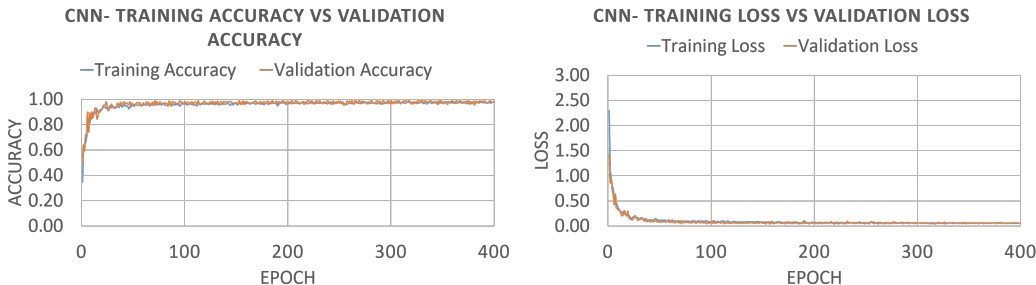

**Figure 18.** Comparison between the loss and accuracy for the CNN algorithm in both training and validation cycles.

Compared to the other algorithms, the GRU started with the highest value of the accuracy, around 0.60, while the others yielded about 30. Moreover, it had the lowest loss. This demonstrated the exceptional learning capacity of the algorithm for the type of problem presented. This was further confirmed by the aggressive overall growth until convergence, with around 35 epochs until the value stabilized. It is also important to point out that the maximum value was achieved in less than 20 epochs, but it took around 40 epochs until the values stabilized. Utilizing just the accuracy and loss criteria, the GRU emerged as the most-suitable algorithm of the ones studied in this work.

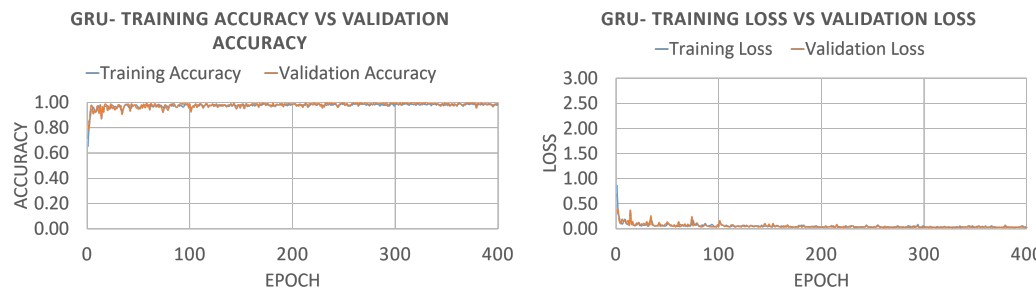

**Figure 19.** Comparison between the loss and accuracy for the GRU algorithm in both training and validation cycles.

The LSTM presented an initial and final accuracy similar to that of the CNN and FNN. The most-striking difference was the low learning rate that the algorithm showed, taking between 150 and 200 epochs to converge. This can be further seen with the erratic initial behavior of this algorithm, which implies that the weight values were being altered, which created an overshooting effect. When only analyzing Figure 20, it can be concluded that the LSTM algorithm was the least suitable for the problem at hand.

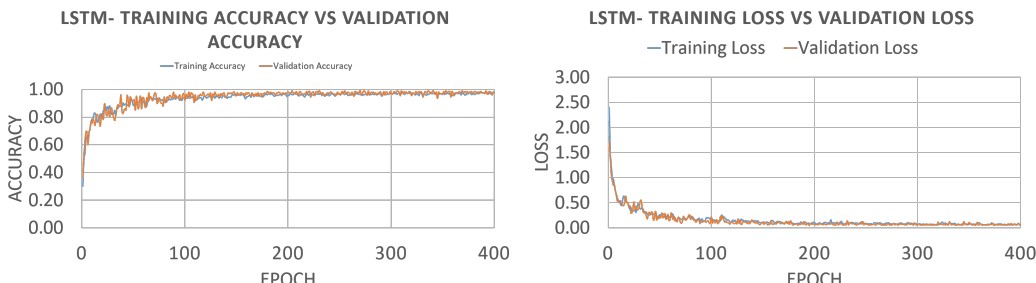

**Figure 20.** Comparison between the loss and accuracy for the LSTM algorithm in both training and validation cycles.

As expected, due to the low layer count of all algorithms, there was no plateau followed by large decreases in the loss. With the plots broadly observed, it is still possible to dive deeper by using the variables listed in Table 2, where a condensed version of the information obtained from the graphs can be seen. These variables are commonly used to describe the performance and represent information that could be seen in a confusion matrix or error matrix:

- The **accuracy score (acc)** is used to evaluate the ratio of correct predictions to the total number of instances. This can be calculated as

$$acc = \frac{tp + tn}{tp + fp + tn + fn}; \tag{15}$$

where $tp$ denotes true positives, $tn$ stands for true negatives, $fp$ represents false positives, and finally, $fn$ corresponds to false negatives.

- The **precision score (p)** is the ratio of correctly predicted positive observations to the total number of positive ones. From a more intuitive perspective, it is the capacity of the algorithm to not label a positive sample as a negative one. It is given by:

$$p = \frac{tp}{tp + fp}; \tag{16}$$

- The **recall score (r)** is the ratio of correctly predicted positive observations in a class, thus from an intuitive perspective, it is the capacity of the algorithm to determine all positive samples. Its value is calculated as:

$$r = \frac{tp}{tp + fn}; \tag{17}$$

- The **F1-score (F1)** is the weighted average of the recall and precision and is commonly more important when there are large imbalances of the data in the class distributions. It is given by:

$$F1 = \frac{2 \times precision \times recall}{precision + recall}. \tag{18}$$

In the case of the tested algorithms, the lowest values can be seen for the CNN, while the highest ones were observed for the FNN and GRU models. Even with these differences, the overall results showed that all algorithms could be used to determine the classes presented in this work. It is worth noting that, when evaluating an SLJ, it is more important that the algorithm does not give false negatives than it generates false positives. This remark makes the recall score more important than the other indices. Indeed, the recall score is directly dependent on the false negatives, while the other criteria use false negatives more indirectly. Using this as a basis, it is possible to see in Table 2 that both LSTM and the CNN presented lower recall values than the FNN and GRU. This makes

both LSTM and the CNN less suitable for the problem at hand and both the FNN and GRU the most-promising ones.

**Table 2.** Algorithms' evaluation criteria.

| | Evaluation Criteria | | | |
|---|---|---|---|---|
| **Algorithm** | **Accuracy** | **Precision** | **Recall** | **F1-Score** |
| FNN | 0.99 | 0.99 | 0.99 | 0.99 |
| CNN | 0.975 | 0.97 | 0.98 | 0.97 |
| GRU | 0.99 | 0.99 | 0.99 | 0.99 |
| LSTM | 0.98 | 0.98 | 0.98 | 0.98 |

It is important to understand that the accuracy, precision, recall, and F1-scores of Expressions (15)–(18) are relevant, but other evaluation criteria need also to be analyzed, such as the number of epochs to reach convergence and the total time taken to achieve the solutions. In Table 3, it is possible to compare the results, which give a more accurate insight into each algorithm. One surprising value is that, although the GRU took longer to calculate for each epoch compared to the CNN and FNN, the total epochs required were fewer, which allowed for the total time to be minimized. It is also possible to see that the GRU and LSTM would largely benefit from a feature extraction protocol, which could reduce the number of features to a few dozens, as large datasets tend to slow calculations and the training process.

**Table 3.** Performance comparison of the algorithms.

| | Epochs and Total Time | | |
|---|---|---|---|
| **Algorithm** | **Epochs to Converge** | **Time to Converge (min)** | **Total Time (min)** |
| FNN | 100 | 7 | 28.0 |
| CNN | 150 | 4.5 | 12.3 |
| GRU | 35 | 15 | 177.5 |
| LSTM | 150 | 89 | 238.3 |

When considering the results summarized in Tables 2 and 3, it is possible to see that, out of all the algorithms, the CNN presented good results, as it was only 2% off the best accuracy and could accomplish this in almost half the time of the next fastest algorithm. At the other end of the spectrum, LSTM took quite a long time to reach the solution and presented the second-lowest value for the accuracy. Finally, the FNN and GRU presented the best accuracy and had relatively acceptable convergence times, making them overall the most-suitable choices.

## 6. Conclusions

This paper proposed an NDT technique based on LW data and artificial intelligence algorithms to detect weak adhesion defects in an SLJ. Four algorithms were trained and tested with simulated LW data from an FEM developed and experimentally validated. The algorithms were assessed and compared. A total of 20 different levels of weak adhesion could be detected, with overall accuracy between 97% and 99%. The results showed the effectiveness of the proposed nondestructive methods to treat the problem, which, until now, could only be solved with classical destructive mechanical testing. Furthermore, the overall training capability was studied, revealing that the training time for each algorithm could be reduced by almost 60% by altering the algorithm's parameters. The LSTM emerged as the least-suitable, taking over 200 epochs to converge, and the GRU the most-applicable, converging in less than 50 epochs. Regarding the training time, the FNN proved to be more appropriate. Therefore, the FNN and GRU presented the best accuracy and showed

relatively acceptable convergence times, making them the most-suitable choices. Further work will expand the technique to increase the resolution for detecting weak adhesion, as well as to localize defects in adhesive joints with a larger and more complex geometry.

**Author Contributions:** Conceptualization, G.M.F.R., L.F.M.D.S. and A.M.L.; methodology, G.M.F.R. and A.M.L.; software, G.M.F.R.; validation, G.M.F.R., L.F.M.D.S., A.M.L. and R.J.C.C.; formal analysis, G.M.F.R., L.F.M.D.S., A.M.L. and R.J.C.C.; investigation, G.M.F.R. and A.M.L.; writing—original draft preparation, G.M.F.R.; writing—review and editing, G.M.F.R., L.F.M.D.S., A.M.L. and R.J.C.C.; supervision, L.F.M.D.S. and A.M.L.; project administration, L.F.M.D.S. and A.M.L. All authors have read and agreed to the published version of the manuscript.

**Funding:** This research received no external funding.

**Informed Consent Statement:** Not applicable.

**Data Availability Statement:** The data presented in this study are available on request from the corresponding author.

**Conflicts of Interest:** The authors declare no conflict of interest.

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
