# Peer review of "Identifying Weak Adhesion in Single-Lap Joints Using Lamb Wave Data and Artificial Intelligence Algorithms"

_applsci, doi:10.3390/app13042642_

Round 1

Reviewer 1 Report

Manuscript ID: applsci-2219504 entitled “Identifying Weak Adhesion in Single Lap Joints Using Lamb Waves Data and Artificial Intelligence Algorithms” for journal of “Applied Sciences” has been reviewed.

- This article is comprehensive, logically organized and contains valuable information.

 However, there are few things need to be corrected and included in the manuscript for better understanding of carried research work to the readers.

+1- The abstract section should be enriched with results obtained (too short).

+2- The novelty of the study should be further explained (introduction section…).

+3-  It will be useful to give general information about composite materials from different studies. (recent studies, 2021-2022).

+4- The resolution of Figure 2, 5, 8, 9, 10, 11, 12, 13 and 16 should be increased. (and magnify)

+5- The expressions in Figure 1 should be corrected.

+6- How did you determine the hidden layers in FNN? (Estimated or based on literature?)

+7- Please, rewrite and check the "constant error carousel" step by step.

+8- The Conclusions section should be enriched with results obtained (too short).

+9- Conclusions section should be enriched a little more. (especially with values)

+10-  More literature studies should be added to the introduction section (DOIs given below).

DOI-1  https://doi.org/10.1515/mt-2020-0024      (info about composites)

DOI-2  https://doi.org/10.26701/ems.923798 (different studies)

---------------------------------------------

*The article will be ready for publication after the specified revisions are made.

**After revision, I would like to review the article again.

------------------------------------------------------

Congratulations to the authors.

I wish the authors success in their future academic studies.

Kind regards.

Author Response

+1- The abstract section should be enriched with results obtained (too short).

  • Thank you. The abstract was further developed and enriched as requested.

+2- The novelty of the study should be further explained (introduction section…).

  • Thank you. The introduction was changed in order to explain more in depth the novelty of the study.
  • Currently, weak adhesion in adhesive joints can only be effectively detected through destructive tests. In this paper, we propose nondestructive testing techniques based on Lamb waves (LW) data and artificial intelligence algorithms. Four algorithms are adjusted and compared in terms of accuracy and training burden. The four methods for LW data processing allow not only to accurately detect up to 20 different levels of weak adhesion in SLJ, but also to choose an efficient training method for the problem at hand. The proposed framework allows the correlation between the methods' parameters, accuracy, and training time, and can be further used with different data and algorithms.

+3-  It will be useful to give general information about composite materials from different studies. (recent studies, 2021-2022).

  • The Introduction section was enriched with recent references on the topic.

+4- The resolution of Figure 2, 5, 8, 9, 10, 11, 12, 13 and 16 should be increased. (and magnify)

  • Thank you. All figures had their size and resolution increased as requested.

+5- The expressions in Figure 1 should be corrected.

  • The expressions in Figure 1 were revised.

+6- How did you determine the hidden layers in FNN? (Estimated or based on literature?)

  • Thank you. The number of hidden layers was tested together with multiple values of kernel and batch sizes, and the ones with smallest training times and highest accuracy were chosen. This can be seen in more detail in Figure 8 and 9 with the various testing steps that were taken into account. We clarified the issue in the revised paper.

+7- Please, rewrite and check the "constant error carousel" step by step.

  • Thank you for the indication, all equations were evaluated and necessary changes made.

+8- The Conclusions section should be enriched with results obtained (too short).

- Thank you. We changed the Conclusion, trying to be more informative.

+9- Conclusions section should be enriched a little more. (especially with values)

- The conclusion was revised and numerical results were included.

+10-  More literature studies should be added to the introduction section (DOIs given below).

DOI-1  https://doi.org/10.1515/mt-2020-0024      (info about composites)

DOI-2  https://doi.org/10.26701/ems.923798 (different studies)

Thank you for the references. The text was further developed and recent studies, included the ones suggested were included.

Reviewer 2 Report

The authors have done a good job. Weak adhesive adhesion may be caused by various factors such as using old glue, poor mixing of two-component glue ingredients, or poor surface preparation. Please describe what glues can be tested by the presented method and what properties.

Author Response

The authors have done a good job. Weak adhesive adhesion may be caused by various factors such as using old glue, poor mixing of two-component glue ingredients, or poor surface preparation. Please describe what glues can be tested by the presented method and what properties.

  • In this work an Epoxy based adhesive is used (Nagase T-836/R-810), therefore the algorithms tested would work for any adhesive with an epoxy base that has similar mechanical properties. This was further clarified in the text.

Reviewer 3 Report

In this paper, nondestructive testing techniques based on Lamb waves data and artificial intelligence algorithms are proposed to detect weak adhesion.

A dataset consisting of simulated Lamb wave time-series extracted from single lap joints subjected to multiple levels of weak adhesion is generated. The raw time-series are pre-processed to standardize the data and allow their use with any dataset size. The processed data are used as input to different artificial intelligence algorithms for their training and testing. The results are used to optimize the methods and to compare their performance. With the proposed framework, it is possible not only to accurately detect up to 20 different levels of weak adhesion, but also to create standardized data and optimal algorithm selection for further work on damage detection and localization in adhesive joints.

In general this paper’s subject matter is well within the journal topic areas, however there are a number of problems and uncertainties that need the authors’ serious attention, and a significant re-write is required before we can assess it again. In this form the paper is not suitable for publication in the Journal of  Biomedical Materials Research Part, B Biomaterials. The reviewer recommends the rejection of paper or the resubmission after major revision. The following are problem areas:

1.      The English in parts of the paper is confusing, leading to uncertainty in understanding the authors’ meaning.

2.      The manuscript has many typos/errors.

3.      (Introduction it’s insufficient a more detail is required.

4.      Results and Discussion part is too weak. The authors just present a Results part and there is no results discussion.

5.      All Figures needs labelling properly. The scale bar for all figures is unclear.

6.      Please provides a high resolution figures to meet the journal requirement

7.      Conclusion: It is too bulky. Make it concise form possibly with some numerical results.

8.      Conclusions must be comprehensive and not written like a report.

9.      It’s very important if you added some references of 2021-2022

Author Response

  1. The English in parts of the paper is confusing, leading to uncertainty in understanding the authors’ meaning.
  • Thank you. The text has been revised to have a more objective and easier to understand document.
  1. The manuscript has many typos/errors.
  • Thank you. The text has been revised to remove the existing typos.
  1. Introduction it’s insufficient a more detail is required.
  • The introduction was changed. We include more details of the process and more references to previous related studies.
  1. Results and Discussion part is too weak. The authors just present a Results part and there is no results discussion.
  • Thank you. The results and discussion were expanded upon.
  1. All Figures needs labelling properly. The scale bar for all figures is unclear.
  • All Figures have their labelling altered as requested, along with the scale bar on the required figures.
  1. Please provides a high resolution figures to meet the journal requirement
  • All figures have been altered to improve resolution.
  1. Conclusion: It is too bulky. Make it concise form possibly with some numerical results.
  • The conclusion was revised and numerical results were included.
  1. Conclusions must be comprehensive and not written like a report.
  • Conclusions were revised.
  1. It’s very important if you added some references of 2021-2022
  • The text was altered to contain recent references.

Round 2

Reviewer 1 Report

Manuscript ID: applsci-2219504 entitled “Identifying Weak Adhesion in Single Lap Joints Using Lamb Waves Data and Artificial Intelligence Algorithms” for journal of “Applied Sciences” has been reviewed.

-Abstract clearly presents objects methods and results.

-Scientific methods are adequately used.

-Terminology is adequate.

-Results are clearly presented.

-Conclusions are logically derived from the data presented.

-Keywords are adequate.

-References are appropriate.

*** Decision: accept

The authors have revised the manuscript carefully and the revised version could be published in the journal.

-----------------------------------------------------

Congratulations to the authors.

I wish the authors success in their future academic studies.

Kind regards.